# HYBRID MEMORY REPLAY:
# BLENDING REAL AND DISTILLED DATA FOR CLASS INCREMENTAL LEARNING

## ABSTRACT

Incremental learning (IL) aims to acquire new knowledge from current tasks while retaining knowledge learned from previous tasks. Replay-based IL methods store a set of exemplars from previous tasks in a buffer and replay them when learning new tasks. However, there is usually a size-limited buffer that cannot store adequate real exemplars to retain the knowledge of previous tasks. In contrast, data distillation (DD) can reduce the exemplar buffer's size, by condensing a large real dataset into a much smaller set of more information-compact synthetic exemplars. Nevertheless, DD's performance gain on IL quickly vanishes as the number of synthetic exemplars grows. To overcome the weaknesses of real-data and synthetic-data buffers, we instead optimize a hybrid memory including both types of data. Specifically, we propose an innovative modification to DD that distills synthetic data from a sliding window of checkpoints in history (rather than checkpoints on multiple training trajectories). Conditioned on the synthetic data, we then optimize the selection of real exemplars to provide complementary improvement to the DD objective. The optimized hybrid memory combines the strengths of synthetic and real exemplars, effectively mitigating catastrophic forgetting in Class IL (CIL) when the buffer size for exemplars is limited. Notably, our method can be seamlessly integrated into most existing replay-based CIL models. Extensive experiments across multiple benchmarks demonstrate that our method significantly outperforms existing replay-based baselines. Our source code is available at https://anonymous.4open.science/r/DD4CIL-510C/.

## 1 INTRODUCTION

Incremental Learning (IL) Wu et al. (2019); Gepperth & Hammer (2016); Douillard et al. (2022); Xie et al. (2022) is an emerging technique that emulates the human ability to continuously acquire new knowledge in an ever-changing world. However, the key challenge is that IL always suffers from *catastrophic forgetting* McCloskey & Cohen (1989), which means the model drastically forgets previously acquired information upon learning new tasks. Thus, how to mitigate catastrophic forgetting in IL remains a significant challenge.

In this work, we primarily focus on a particularly challenging scenario known as class incremental learning (CIL) Douillard et al. (2020); Zhu et al. (2021); Wang et al. (2022b;a). CIL aims to learn new and disjoint classes across successive task phases and then accurately predicts all classes observed thus far in each phase, without prior knowledge of task identities. A common approach to enhance CIL performance is the use of replay, which stores a subset of exemplars from old tasks and revisits them when learning new tasks. Storing real exemplars can significantly enhance the performance of CIL Rebuffi et al. (2017); Chaudhry et al. (2019); however, the performance will be affected by the limitation of the exemplar buffer size. To decrease the dataset size, recent studies have explored data distillation (DD) Loo et al. (2022); Du et al. (2023) to distill compact *synthetic exemplars* from real datasets. However, Yu et al. (2023) indicates that the effectiveness of synthetic exemplars declines as their quantity increases. Motivated by these findings, we aim to develop a new hybrid memory that combines the advantages of both real and synthetic exemplars for replay-based CIL methods.

In this paper, we introduce a novel hybrid memory that optimizes a limited number of both synthetic and real exemplars for replay-based CIL methods. A key challenge is enabling these limited exemplars to capture adequate information from data samples of previous tasks. To address this challenge, we aim to introduce data distillation (DD) into CIL, which condenses large datasets into a small set of representative samples. However, classical DD techniques are not directly applicable to replay-based CIL due to the requirement of checkpoints on multiple training trajectories. To address this challenge, we first propose an innovative Continual Data Distillation (CDD) method that adapts DD for replay-based CIL, extracting informative synthetic exemplars from previous tasks' datasets. Interestingly, experiments in Sec. 3 across different limited exemplar buffer sizes reveal that while memory relying solely on synthetic exemplars initially outperforms memory using only real exemplars, its performance becomes less effective than the memory only with real exemplars as the buffer size increases. To address this issue, we further develop conditional real data selection that chooses optimal real samples to complement synthetic data. As a result, the interplay between selected real ex-

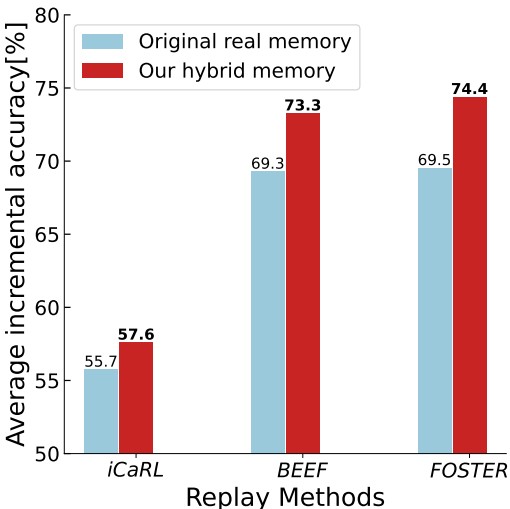

Figure 1: Performance comparison between the real memory as used by the original CIL methods and our hybrid memory across multiple baselines: iCaRL Rebuffi et al. (2017), BEEF Wang et al. (2022a), and FOSTER Wang et al. (2022b), all with the same exemplar buffer size on CIFAR-100 Krizhevsky & Hinton (2009).

emplars and synthetic data enhances model performance within a limited exemplar buffer size. As illustrated in Fig. 1, our proposed hybrid memory significantly outperforms the original real exemplars across various replay-based CIL methods. Finally, in Sec. 5, we evaluate our approach by integrating it into several existing replay-based CIL models and compare their performance against baseline models. The results validate that our hybrid memory consistently leads to superior performance.

**Our contributions are four-fold:** 1) to our best knowledge, we are the first to synergize limited synthetic and real exemplars to boost replay-based CIL performance; 2) we develop a novel conditional real data selection that optimally chooses real exemplars, which can complement the synthetic data effectively; 3) our approach can be inserted into many existing replay-based CIL models to improve their performance; and 4) extensive experiments demonstrate that our method can significantly outperform existing replay-based CIL approaches with a limited number of saved exemplars.

## 2 RELATED WORK

**Class Incremental Learning.** Existing works of CIL can be categorized into three types: regularization-based methods, architecture-based methods, and replay-based methods.

(i) **Regularization-based methods** seek to mitigate catastrophic forgetting by incorporating explicit regularization terms to balance the knowledge from old and new tasks Li & Hoiem (2017); Kirkpatrick et al. (2017); Lee et al. (2017); Liu et al. (2018); Ahn et al. (2019). However, only using regularization-based methods often proves inadequate for preventing catastrophic forgetting. Therefore, these methods often collaborate with the replay-based methods to boost performance. (ii) **Architecture-based methods** try to mitigate catastrophic forgetting by either dynamically expanding the model's parameters to accommodate new tasks Mallya & Lazebnik (2018); Ahn et al. (2019); Douillard et al. (2022) or by learning new tasks in parallel using task-specific sub-networks or modules Rusu et al. (2016); Fernando et al. (2017); Henning et al. (2021). However, these approaches often require substantial memory to store the expanded network or multiple sub-networks and typically depend on task identities to determine the appropriate parameters or sub-networks for a given task. Consequently, their applicability is significantly limited. (iii) **Replay-based methods** are designed to mitigate catastrophic forgetting by preserving a small subset of old task exemplars for replay while learning new tasks. One line of these methods Rebuffi et al. (2017); Douillard et al. (2020); Wang et al. (2022b;a) maintains a limited memory for storing real exemplars from old tasks. However, due to data

privacy concerns and inherent memory constraints, the number of real exemplars stored is typically small, significantly affecting performance in applications. Another line Shin et al. (2017); Ostapenko et al. (2019); Lesort et al. (2019); Rios & Itti (2018) employs generative models, such as Variational Autoencoders (VAE) and Generative Adversarial Networks (GAN), to create synthetic exemplars of old tasks for replay. However, these methods often face challenges with label inconsistency during continual learning Ayub & Wagner (2020); Ostapenko et al. (2019), and is closely related to continual learning of generative models themselves Wang et al. (2023). In contrast, this paper aims to condense dataset information into a few synthetic exemplars without training additional models.

**Data Distillation.** Data distillation (DD) aims to condense the information from a large-scale dataset into a significantly smaller subset. This process ensures that the performance of a model trained on the distilled subset is comparable to that of a model trained on the original dataset. Existing DD methods can be grouped into three categories based on the optimization objectives: *performance matching* Loo et al. (2022); Zhou et al. (2022), *parameter matching* Cazenavette et al. (2022); Zhao & Bilen (2021); Du et al. (2023), and *distribution matching* Zhao & Bilen (2023); Sajedi et al. (2023). While a few studies Zhao & Bilen (2023); Sajedi et al. (2023) have applied DD to a simple specific IL application like GDumb Prabhu et al. (2020), they cannot be directly applied to the general replay-based CIL methods. This is because traditional DD techniques require training *a set of models* randomly sampled from an initialization distribution. However, for typical replay-based CIL, the initial model parameters for each task are derived from the weights trained on the previous task. Thus, directly applying existing DD techniques to CIL introduces significant computational costs, as it requires training an additional set of models to get multiple trajectories. To address this issue, we introduce the Continual DD (CDD) technique to suit replay-based CIL.

## 3 WHY HYBRID MEMORY?

In this section, we mainly introduce the rationale behind the need for hybrid memory in CIL by comparing real, synthetic, and bybrid memory.

To address catastrophic forgetting in CIL, general replay-based methods attempt to store real exemplars from previous tasks in a limited exemplar buffer. Intuitively, saving high-quality exemplars from old tasks can enhance the performance of replay-based CIL. However, a size-limited exemplar buffer cannot store adequate real exemplars to retain the knowledge of previous tasks. To address this challenge, we introduce a Continual Data Distillation (CDD) to compress important information from each task's dataset into a limited number of synthetic exemplars, replacing the need to store real exemplars in the buffer.

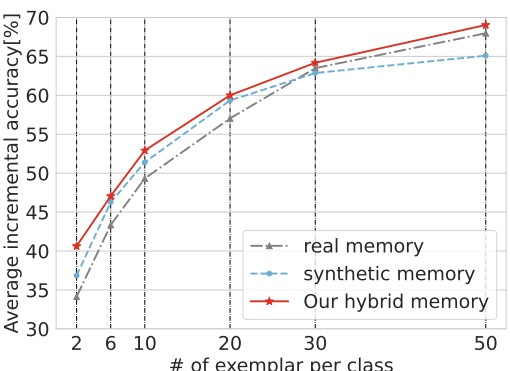

Figure 2: Performance evaluation of iCaRL Rebuffi et al. (2017) using different exemplar buffer sizes for real memory, synthetic memory, and our hybrid memory. "real memory" refers to buffers containing only real exemplars selected by iCaRL. "synthetic memory" contains only synthetic exemplars generated by CDD.

However, synthetic exemplars alone are not always sufficient. As their number increases, performance tends to plateau due to the inevitable information loss that occurs during the distillation process. As shown in Fig. 2 and Appendix E, we evaluate various replay-based CIL methods with buffers containing either only real exemplars or only synthetic exemplars generated by CDD across different buffer sizes on CIFAR-100 Krizhevsky & Hinton (2009). We can observe that synthetic memory performs better than real memory when the buffer size is small, suggesting that synthetic exemplars capture more information per exemplar when storage is constrained. However, as the buffer size increases, synthetic exemplars become less effective than real exemplars, likely due to the inherent important information loss in the distillation process. Motivated by these observations, we propose to construct a hybrid memory that combines the conditionally selected real exemplars with the synthetic ones for complementing each other. In the following Sec. 4, we will detail the proposed method.

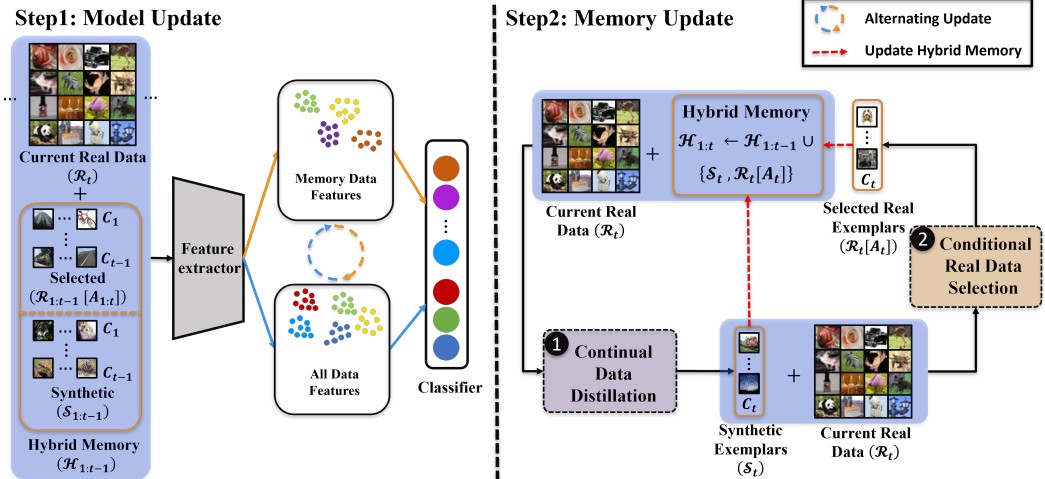

Figure 3: The framework of the proposed hybrid memory system for replay-based CIL. We first leverage the current real data with the hybrid memory for former classes to update the model. Then we use Continual Data Distillation (❶) to extract synthetic exemplars and conditional real data selection (❷) to choose optimal exemplars conditioned on synthetic data. Finally, the synthetic exemplars and selected real exemplars are combined to update the hybrid memory.

# 4 PROPOSED HYBRID MEMORY

In this section, we first present the problem of using hybrid memory in replay-based CIL. Then we detail the proposed hybrid memory to mitigate the catastrophic forgetting in CIL.

## 4.1 PROBLEM FORMULATION

This work aims to build a high-quality hybrid memory to mitigate the catastrophic forgetting problem in CIL, which involves sequentially learning distinct tasks, each consisting of disjoint groups of classes. Specifically, the real dataset for task-$t$ is denoted as $\mathcal{R}_t \triangleq \{(x_i, y_i)\}_{i=1}^{n_t}$, where $y_i$ belongs to the label space $C_t$ of task-$t$, and $n_t$ represents the number of instances. Unlike traditional replay-based methods that store only limited real exemplars, our approach maintains a hybrid memory with a limited number of hybrid exemplars from learned tasks' classes. The hybrid memory is represented as $\mathcal{H}_{1:t-1} \triangleq \bigcup_{i=1}^{t-1} \mathcal{H}_i$, where $\mathcal{H}_t \triangleq \{\mathcal{S}_t, \mathcal{R}_t[A_t]\}$ represents the hybrid memory for task-$t$. Here, $\mathcal{S}_t$ comprises synthetic exemplars for the classes in task-$t$, and $A_t \subseteq V_t$ ($V_t = \{1, 2, \cdots, |\mathcal{R}_t|\}$ is the ground set) indexes a subset of real samples selected from $\mathcal{R}_t$. The model $f(x; \theta)$ trained on $\mathcal{H}_{1:t-1} \cup \mathcal{R}_t$ is required to predict probabilities for all former classes, including the current task-$t$, denoted as $C_{1:t} \triangleq \bigcup_{i=1}^{t} C_i$, for any given input $x$. **We define the important notations used in this work in Appendix A**.

## 4.2 OVERVIEW OF THE PROPOSED HYBRID MEMORY

Fig. 3 presents the framework of the proposed hybrid memory system for CIL, comprising two key components: (i) model update and (ii) memory update. The model update refines the CIL model by leveraging a hybrid memory composed of both synthetic and real exemplars. Instead of trivially merging synthetic and real data, our memory update jointly optimizes the two by selecting real exemplars based on the optimized synthetic data. As the memory update is central to our method, we first elaborate on this component below.

## 4.3 MEMORY UPDATE

According to Section 3, we propose to obtain an optimal hybrid memory $\mathcal{H}_t^\star$ for task-$t$ by optimizing the following objective,

$$\mathcal{H}_t^\star \in \arg\min_{\mathcal{H}_t} \mathcal{L}(\mathcal{H}_t, \mathcal{R}_t), \ \ \mathcal{H}_t \triangleq \{\mathcal{S}_t; \mathcal{R}_t[A_t]\}, \tag{1}$$

where $\mathcal{L}(\cdot, \cdot)$ measures the distance between the hybrid memory $\mathcal{H}_t$ and the real dataset $\mathcal{R}_t$. The hybrid memory minimizing $\mathcal{L}(\cdot, \cdot)$ is expected to achieve comparable performance as $\mathcal{R}_t$ when used to train the model.

However, jointly optimizing both the synthetic exemplars $\mathcal{S}_t$ and the subset $A_t$ of real exemplars at the same time is challenging, as the optimization of $\mathcal{S}_t$ is a continuous process while selecting real exemplars $\mathcal{R}_t[A_t]$ requires combinatorial optimization. To address this problem, we adopt an alternating optimization strategy to optimize the hybrid memory, as illustrated on the right side of Fig. 3. Below, we detail our method for constructing an effective hybrid memory.

❶ **Continual Data Distillation (CDD).** First, we aim to optimize the synthetic exemplars $\mathcal{S}_t^\star$ for hybrid memory. Traditional DD methods can condense dataset information into a few synthetic exemplars, but they cannot be directly applied to replay-based CIL methods. This is because existing DD methods distill synthetic data from multiple training trajectories (with different random initialization) while most CIL methods only produce one trajectory per task. To address this issue, we introduce a novel Continual Data Distillation (CDD) technique, which distills $\mathcal{S}_t$ from checkpoints on task-$t$'s training trajectory $\theta_{1:N}$, where $N$ represents the total number of cached checkpoints $\theta$ in the history of learning task-$t$. We do not need multiple trajectories because the replay of $\mathcal{S}_t$ in the next task will only start from $\theta_N$ instead of random initializations.

Specifically, to distill synthetic exemplars from the real dataset (❶) after training the model on task-$t$, we will minimize the distance ($\mathcal{L}(\cdot, \cdot)$ in Eq. 1) between synthetic exemplars and the real data from task-$t$. To effectively solve this equation, we propose CDD to adapt the original DD optimization objective into CIL below,

$$\mathcal{S}_t^\star \in \underset{\mathcal{S}_t}{\arg\min} \, \mathcal{L}(\mathcal{S}_t, \mathcal{R}_t) = \underset{\theta \sim \theta_{1:N}}{\mathbb{E}} \left[ \ell(\mathcal{S}_t, \mathcal{R}_t; \theta) \right], \tag{2}$$

where $\ell(\cdot, \cdot; \cdot)$ in Eq. 2 is a DD objective from any existing DD methods. It aims to measure the distance between the synthetic exemplars $\mathcal{S}_t$ and the real data $\mathcal{R}_t$ using the sampled model parameters $\theta$. **In Appendix D, we present a detailed description of several optimization objectives for CDD in CIL.**

To keep a constant and smaller memory of model checkpoints, we can instead apply a sliding window version of CDD of window size $\tau \ll N$, i.e., updating $\mathcal{S}_t$ only based on the most recent $\tau$ checkpoints. Specifically, after each epoch-$j+1$ of task-$t$, we update the synthetic data $\mathcal{S}_t$ as $\mathcal{S}_{t,j+1}$ below.

$$\mathcal{S}_{t,j+1} = \mathcal{S}_{t,j} - \eta \nabla_{\mathcal{S}_{t,j}} \left( \underset{\theta \sim \theta_{j-\tau+2:j+1}}{\mathbb{E}} \left[ \ell(\mathcal{S}_{t,j}, \mathcal{R}_t; \theta) \right] \right), \tag{3}$$

where $\eta$ is the learning rate for updating the synthetic exemplars $\mathcal{S}_{t,j+1}$. Hence, we only need to store $\tau$ checkpoints in any step but the final $\mathcal{S}_t$ at the end of each task is a result of DD on all the $N$ checkpoints. The final $\mathcal{S}_t^\star \leftarrow \mathcal{S}_{t,N}$.

❷ **Conditional Real Data Selection**. While synthetic exemplars effectively capture rich information from previous tasks, their impact tends to plateau as their quantity increases (see Sec. 3). To overcome this limitation, we select a subset of real data complementary to $\mathcal{S}_t^*$ at the end of each task $t$ to further reduce the gap between the memory $\mathcal{H}_t$ and $\mathcal{R}_t$ (refer to ❷ in Fig. 3). Guided by Eq. 1, the objective is to find an optimal subset $A_t^\star \subseteq V_t$ such that the selected real data $\mathcal{R}_t[A_t^\star]$, in conjunction with the distilled synthetic exemplars $\mathcal{S}_t^\star$, minimize the DD objective below:

$$A_t^\star \in \underset{A_t \subseteq V_t, |A_t| \leq k}{\arg\min} \, \ell([\mathcal{R}_t[A_t]; \mathcal{S}_t^\star], \mathcal{R}_t; \theta_t), \tag{4}$$

where $\theta_t$ denotes the model parameters trained on $\mathcal{R}_t \cup \mathcal{H}_{1:t-1}$, task-$t$'s real data $\mathcal{R}_t$ combined with the hybrid memory $\mathcal{H}_{1:t-1}$ for the former classes. The DD optimization objective $\ell(\cdot, \cdot; \cdot)$ is kept consistent with the one in Eq. 2. Since the conditional subset selection problem in Eq. 4 is an NP-hard combinatorial optimization problem, we apply the greedy algorithm to approximately optimize the subset $A_t$.

**Algorithm.** To select some ideal real exemplars, we employ a greedy algorithm within our selection. This approach involves iteratively selecting the locally optimal real exemplars from the current task, guided by the objective in Eq. 4. The core idea is to approximate a globally optimal selection, designated as $\mathcal{R}_t[A_t^\star]$, through these successive local optimizations. **The detailed algorithm is summarized in Appendix B.**

**Theoretical Analysis.** We also theoretically analyze the performance of the model trained on the proposed hybrid memory, showing that it can achieve performance comparable to a model trained on the real dataset for all tasks in CIL. We begin our analysis with learning CIL tasks jointly, which serves as the performance benchmark for CIL. In this scenario, the training data for task-$t$ is written as $\mathcal{R}_{1:t} \triangleq \bigcup_{i=1}^{t-1} \mathcal{R}_i \cup \mathcal{R}_t$ and the model for task-$t$ can be obtained by the maximum likelihood estimation below

$$\theta_{\mathcal{R}_{1:t}} = \arg\max_{\theta}(\log P(\mathcal{R}_{1:t-1}|\theta) + \log P(\mathcal{R}_t|\theta)). \tag{5}$$

Given that the absence of $\mathcal{R}_{1:t-1}$ leads to catastrophic forgetting in CIL, our objective is to construct a smaller hybrid memory, $\mathcal{H}_{1:t-1}$, with limited exemplars that enable the network to achieve performance comparable to that obtained when trained on $\mathcal{R}_{1:t-1}$.

**Definition 1.** *The optimal model trained on task-$t$'s hybrid memory is* $\theta_{\mathcal{H}_t} = \arg\max_{\theta} \log P(\mathcal{H}_t|\theta)$.

**Definition 2.** *The optimal model trained on task-$t$'s real data is* $\theta_{\mathcal{R}_t} = \arg\max_{\theta} \log P(\mathcal{R}_t|\theta)$.

**Definition 3.** *The optimal model trained on the hybrid memory of all prior tasks from 1 to $t$ is denoted by* $\theta_{\mathcal{H}_{1:t}} = \arg\max_{\theta} h_t(\theta) = \arg\max_{\theta} \log P(\mathcal{H}_{1:t}|\theta)$.

**Definition 4.** *The optimal model trained on the real data of all tasks from 1 to $t$ is defined as* $\theta_{\mathcal{R}_{1:t}} = \arg\max_{\theta} r_t(\theta) = \arg\max_{\theta} \log P(\mathcal{R}_{1:t}|\theta)$.

**Assumption 1.** *Based on our objective of the proposed hybrid memory, we assume that the model trained on the hybrid memory for one task can achieve comparable performance to that of the model trained solely on real data for the same task, we can express this formally as* $\exists \epsilon_t \in [0, 1), \log P(\mathcal{R}_{t+1}|\theta_{\mathcal{H}_{t+1}}) \geq (1 - \epsilon_t) \log P(\mathcal{R}_{t+1}|\theta_{\mathcal{R}_{t+1}})$.

**Assumption 2.** *Assume that the performance of the model trained on the hybrid memory of all prior tasks till task-$t$ and the performance of the optimal model trained on the hybrid memory of task-$t + 1$ can be bonded as* $\exists \rho, \rho * r_{t+1}(\theta_{\mathcal{R}_{1:t+1}}) \geq r_t(\theta_{\mathcal{H}_{1:t}}) + \log P(\mathcal{R}_{t+1}|\theta_{\mathcal{H}_{t+1}})$.

**Theorem 1** (Performance Approximation). *Based on the above Assumptions 1 and 2, when* $\epsilon_{t+1} \geq \frac{\rho}{1-\epsilon_t}$, *we can derive that the model trained on the hybrid memory of all previous tasks achieves performance comparable to that of the model trained on the real dataset of all previous tasks,*

$$\log P(\mathcal{R}_{1:t}|\theta_{\mathcal{H}_{1:t}}) \geq (1 - \epsilon_t) \log P(\mathcal{R}_{1:t}|\theta_{\mathcal{R}_{1:t}}). \tag{6}$$

**We provide the detailed proof of Theorem. 1 in Appendix. C.**

**Remark.** *According to Theorem 1, when $\epsilon_t$ and $\rho$ are small, Eq.6 is well bounded. In other words, if the proposed hybrid memory is optimized using $\mathcal{L}(\cdot, \cdot)$ in Eq.1 for each task in CIL, the performance of the model trained on the hybrid memory across all tasks can achieve results comparable to those of a model trained on a real dataset for all tasks.*

## 4.4 MODEL UPDATE

Next, we describe the model update based on the updated hybrid memory. After optimizing the hybrid memory $\mathcal{H}_t$ for the task-$t$, the hybrid memory is then updated as follows:

$$\mathcal{H}_{1:t} \leftarrow \mathcal{H}_{1:t-1} \cup \{\mathcal{S}_t^\star, \mathcal{R}_t[A_t^\star]\}. \tag{7}$$

We then combine the updated hybrid memory, $\mathcal{R}_{t+1}$, which contains data from previous tasks, with the real exemplars, $\mathcal{R}_{t+1}$, to train the CIL model for task-$t + 1$. As illustrated on the left side of Fig.3, to achieve the optimal model trained on the hybrid memory of all prior tasks, i.e., $\theta_{1:t}^H$ in Eq. 6, we adopt an alternating model update strategy based on the data and hybrid memory of task-$t$.

Finally, we detail the training pipeline for task-$t$ in Algorithm 1.

## 5 EXPERIMENTS

In this section, we first compare the performance of the proposed hybrid memory with the replay-based baselines in CIL. We then demonstrate that the integration of the hybrid memory can enhance

---

**Algorithm 1:** Training with Hybrid Memory on Task-$t$

---

**input**   :Epochs: $N$; sliding window: $\tau$; real data for classes $C_t$: $\mathcal{R}_t$; number of synthetic exemplars per class: $k$; hybrid memory from previous tasks: $\mathcal{H}_{1:t-1}$

1  Randomly select $k$ samples per class from $\mathcal{R}_t$ as $\mathcal{S}_{t,0}$;
2  **for** $j \in \{1, \ldots, N\}$ **do**
3      update the model as $\theta_j$ on $\mathcal{R}_t \cup \mathcal{H}_{1:t-1}$ using a specified CIL objective and cache $\theta_j$;
4      **if** $j > \tau$ **then**
5          remove the cached checkpoint $\theta_{j-\tau}$;
6          update synthetic data $\mathcal{S}_{t,j}$ using Eq. 3;
7      **else**
8          $\mathcal{S}_{t,j} \leftarrow \mathcal{S}_{t,j-1}$;
9  $\mathcal{S}_t \leftarrow \mathcal{S}_{t,N}$;
10  optimize $A_t$ in Eq. 4 using Algorithm 2 to obtain the selected real exemplars as $\mathcal{R}_t[A_t]$;
  **output**   :Hybrid Memory $\mathcal{H}_{1:t} \leftarrow \mathcal{H}_{1:t-1} \cup \{\mathcal{S}_t, \mathcal{R}_t[A_t]\}$

---

the performance of some existing replay-based CIL methods. Additionally, we investigate the ratio of synthetic exemplars within the hybrid memory. Finally, we examine the effectiveness of our conditional real data selection in choosing ideal real exemplars.

**Datasets.** We evaluate the proposed method on two commonly used benchmarks for class incremental learning (CIL), CIFAR-100 Krizhevsky & Hinton (2009) and TinyImageNet Yao & Miller (2015). ***CIFAR-100:*** CIFAR-100 contains 100 classes, the whole dataset includes $50,000$ training images with $500$ images per class and $10,000$ test images with $100$ images per class. ***TinyImageNet:*** TinyImageNet contains 200 classes, the whole dataset includes $100,000$ training images with $500$ images per class and $10,000$ test images with $50$ images per class.

**Protocol.** To evaluate the performance of our method, we apply two commonly used protocols Zhou et al. (2023) for both datasets. 1) **zero-base:** For the zero-base setting, we split the whole classes into 5, and 10 tasks (5, and 10 phases) evenly to train the network incrementally. 2) **half-base:** For the half-base setting, we train half of the whole classes as the first task and then split the rest half of the classes into 5, and 10 tasks (5, and 10 phases) evenly to train the network incrementally. To ensure a fair comparison across different replay-based methods, we maintain a fixed memory capacity that stores 20 exemplars per class for training.

**Baselines.** We compare some replay-based methods that incorporate our proposed hybrid memory with other established replay-based approaches, including iCaRL Rebuffi et al. (2017), BiC Wu et al. (2019), WA Zhao et al. (2020), PODNet Douillard et al. (2020), FOSTER Wang et al. (2022b), and BEEF Wang et al. (2022a). We assess the performance of these methods using two commonly used metrics in CIL: Average Accuracy (AA) Chaudhry et al. (2018), which represents the overall performance at a given moment, and Average Incremental Accuracy (AIA) Rebuffi et al. (2017), which captures historical performance variations.

**Experiments Setting.** In our experiments, we insert our hybrid memory into various replay-based CIL methods: iCaRL Rebuffi et al. (2017), FOSTER Wang et al. (2022b), and BEEF Wang et al. (2022a). All methods are implemented with PyTorch Paszke et al. (2017) and PyCIL Zhou et al. (2021), which are well-regarded tools for CIL. We use ResNet-18 He et al. (2016) as the feature extractor for all methods, with a uniform batch size of $B = 128$. For iCaRL, we employ the SGD optimizer Ruder (2016), with a momentum of 0.9 and a weight decay of 2e-4. The model is trained for 170 epochs, starting with an initial learning rate of 0.01, which is then reduced by a factor of 0.1 at the 80th and 120th epochs. For FOSTER Wang et al. (2022b), we use SGD with a momentum of 0.9. The weight decay is set at 5e-4 during the boosting phases and 0 during the compression phase. The model is trained for 170 epochs during the boosting phases and 130 epochs during the compression phase. The learning rate starts at 0.01 and follows a cosine annealing schedule that decays to zero across the specified epochs. For BEEF Wang et al. (2022a), we adopt SGD with a momentum of 0.9. The weight decay is 5e-4 during the expansion phase and 0 during the fusion phase. The model is trained for 170 epochs in the expansion phase and 60 in the fusion phase, with an initial learning rate of 0.01, decaying to zero according to a cosine annealing schedule. The detailed hyper-parameter settings for FOSTER and BEEF can be found in the original papers. For the specific data distillation

algorithm for CDD, we adapt DM Zhao & Bilen (2023), a distribution matching DD method, as the objective $\ell(\cdot, \cdot; \cdot)$ in Eq. 2. For the window size $\tau$ in the sliding window version CDD, we use $\tau = 4$. For DM, we use the SGD optimizer with a momentum of 0.5 to learn the synthetic exemplars. The learning rate $\eta$ in Eq. 3 is set to 0.1 for CIFAR-100 and 1 for TinyImageNet. For both datasets, we train the synthetic exemplars for 10,000 iterations. All experiment results are averaged over three runs. We run all experiments on a single NVIDIA RTX A6000 GPU with 48GB of graphic memory.

## 5.1 MAIN RESULTS

First, we compare the performance of the proposed hybrid memory with various replay-based CIL methods on CIFAR-100 and TinyImageNet datasets under zero-base settings with 5 and 10 phases and half-base settings with 5 and 10 phases, storing 20 exemplars per class. To demonstrate the effectiveness and applicability of our approach, we integrate it into three replay-based CIL models: the classical method iCaRL, and the recent methods FOSTER and BEEF. In this experiment, we adopt the DM as our CDD technique to extract synthetic exemplars for the hybrid memory, denoted by $H$. Our hybrid memory comprises 10 synthetic exemplars generated by CDD and 10 real exemplars selected through the proposed conditional real data selection.

Tables 1 and 2 show the comparison of the performance achieved by iCaRL, BEEF, and FOSTER using hybrid memory against different baselines under two different settings. The experimental results are averaged over three runs. Regarding the **zero-base setting**, as shown in Table 1, the FOSTER integrated with our hybrid memory, which achieves the best performance among our three methods in AIA and the last average accuracy (LAA), outperforms the best baseline FOSTER by a large margin on CIFAR-100 under both 5-phase and 10-phase settings. Likewise, the FOSTER with our method achieves higher AIA than the best baseline FOSTER on TinyImageNet for the same settings and also performs the best in LAA on TinyImageNet under the 10-phase setting.

For the **half-base setting**, as shown in Table 2, the BEEF incorporated with our method surpasses the best baseline (FOSTER) with a large margin in AIA and LAA on CIFAR-100 under the 5-phase setting, and on TinyImageNet under both 5-phase and 10-phase settings. Similarly, the FOSTER integrated with the proposed hybrid memory outperforms the best baseline FOSTER in AIA and LAA on CIFAR-100 under the 10-phase setting.

The reason why our method performs well is that the optimized hybrid memory, combining the strengths of synthetic and real exemplars, can retain more useful knowledge of previous tasks. In sum, the experimental results demonstrate the effectiveness of our hybrid memory in improving CIL performance. ***Due to limited space, we present more detailed results in Appendix F***.

Table 1: AIA (average incremental accuracy) and LAA (the last task's average accuracy) of the three replay-based methods with the proposed hybrid memory and baselines on both CIFAR-100 and TinyImageNet under **zero-base** 5 and 10 phases with 20 exemplars per class. "***w H***" denotes using our proposed hybrid memory with distribution matching (DM).

| Method (20 exemplars per class) | CIFAR-AIA [%]↑ | | CIFAR-LAA [%]↑ | | Tiny-AIA [%]↑ | | Tiny-LAA [%]↑ | |
|---|---|---|---|---|---|---|---|---|
| | b0-5 | b0-10 | b0-5 | b0-10 | b0-5 | b0-10 | b0-5 | b0-10 |
| BiC Wu et al. (2019) | 66.57 | 56.34 | 51.90 | 33.33 | 60.20 | 42.12 | 47.41 | 18.22 |
| WA Zhao et al. (2020) | 68.02 | 64.44 | 57.84 | 50.57 | 60.21 | 45.92 | 47.96 | 30.94 |
| PODNet Douillard et al. (2020) | 64.87 | 52.07 | 49.97 | 37.14 | 49.33 | 39.77 | 30.33 | 22.21 |
| iCaRL Rebuffi et al. (2017) | 63.26 | 57.03 | 48.70 | 44.44 | 56.79 | 47.15 | 38.55 | 30.29 |
| BEEF Wang et al. (2022a) | 75.21 | 67.73 | 68.17 | 53.49 | 69.02 | 62.98 | 60.11 | 50.96 |
| FOSTER Wang et al. (2022b) | 78.15 | 75.00 | 70.76 | 65.12 | 68.60 | 65.37 | 57.60 | 52.70 |
| iCaRL *w H* (Ours) | 69.47 | 60.00 | 56.21 | 45.23 | 62.97 | 50.78 | 47.44 | 34.42 |
| BEEF *w H* (Ours) | 76.66 | 71.04 | 68.48 | 59.62 | 69.54 | 64.06 | **62.91** | 52.13 |
| FOSTER *w H* (Ours) | **78.79** | **76.01** | **70.93** | **66.63** | **70.28** | **68.55** | 60.24 | **56.93** |

## 5.2 EFFECTIVENESS OF HYBRID MEMORY

In this section, we demonstrate the overall effectiveness of the proposed hybrid memory in enhancing the performance of some existing replay-based CIL methods. In this experiment, we adapt the original DM objective in CDD, denoted as $\ell(\cdot, \cdot; \cdot)$ in Eq. 2, to generate synthetic exemplars, which make up the synthetic memory ($S$). We then conditionally select the optimal real exemplars based on these synthetic exemplars to construct the hybrid memory ($H$). To validate its general effectiveness, we

Table 2: AIA and LAA of the three replay-based methods with the proposed hybrid memory and baselines on both CIFAR-100 and TinyImageNet under **half-base** (train half of the whole classes as the first task) 5 and 10 phases with 20 exemplars per class. "*w H*" denotes using our proposed hybrid memory with distribution matching (DM).

| Method (20 exemplars per class) | CIFAR-AIA [%]↑ | | CIFAR-LAA [%]↑ | | Tiny-AIA [%]↑ | | Tiny-LAA [%]↑ | |
|---|---|---|---|---|---|---|---|---|
| | half-5 | half-10 | half-5 | half-10 | half-5 | half-10 | half-5 | half-10 |
| BiC Wu et al. (2019) | 64.91 | 58.11 | 49.48 | 41.48 | 51.48 | 44.29 | 36.41 | 25.98 |
| WA Zhao et al. (2020) | 71.90 | 68.08 | 63.34 | 56.30 | 52.55 | 45.32 | 38.45 | 29.07 |
| PODNet Douillard et al. (2020) | 64.84 | 58.81 | 54.09 | 47.46 | 41.84 | 33.05 | 29.80 | 22.88 |
| iCaRL Rebuffi et al. (2017) | 61.91 | 55.73 | 45.62 | 40.87 | 44.38 | 33.02 | 28.86 | 17.50 |
| BEEF Wang et al. (2022a) | 68.57 | 69.27 | 61.56 | 60.90 | 58.39 | 54.69 | 51.52 | 48.05 |
| FOSTER Wang et al. (2022b) | 73.34 | 69.51 | 65.80 | 60.30 | 60.94 | 54.67 | 52.10 | 43.60 |
| iCaRL *w H* (Ours) | 64.13 | 57.58 | 51.02 | 41.60 | 49.16 | 34.91 | 33.23 | 19.99 |
| BEEF *w H* (Ours) | **76.31** | 73.25 | **70.52** | 61.80 | **63.76** | **60.76** | **56.89** | **52.27** |
| FOSTER *w H* (Ours) | 75.70 | **74.37** | 68.53 | **64.95** | 63.60 | 58.57 | 55.67 | 48.73 |

integrate both the synthetic memory and hybrid memory into three replay-based CIL methods: iCaRL, FOSTER, and BEEF. We evaluate the performance of these methods on CIFAR-100 under two configurations: zero-base (5 and 10 phases) and half-base (5 and 10 phases), using 20 exemplars per class. The performance comparison between the original methods and those using our hybrid memory, as shown in Table 3, indicates that integrating hybrid memory significantly enhances performance across all three replay-based CIL methods. Especially, under the half-base setting, our hybrid memory improves iCaRL by 2.22% and 1.85%, BEEF by 7.74% and 3.98%, and FOSTER by 2.36% and 4.86% in AIA for the 5-phase and 10-phase settings. Comparing the performance of the three methods integrated with our hybrid memory (*w H*) against the original methods with real memory and with synthetic memory (*w S*), our hybrid memory consistently outperforms both the synthetic and original real memory across different CIL methods and settings, all while maintaining the same exemplar buffer size.

Table 3: AIA and LAA of the three replay-based methods with the proposed hybrid memory, synthetic exemplars, and baselines on CIFAR-100 under **zero-base** and **half-base** 5 and 10 phases with 20 exemplars per class. "*w S*" denotes using the synthetic memory achieved by CDD with distribution matching (DM).

| Method (20 exemplars per class) | CIFAR-AIA [%]↑ | | CIFAR-LAA [%]↑ | | CIFAR-AIA [%]↑ | | CIFAR-LAA [%]↑ | |
|---|---|---|---|---|---|---|---|---|
| | b0-5 | b0-10 | b0-5 | b0-10 | half-5 | half-10 | half-5 | half-10 |
| iCaRL | 63.26 | 57.03 | 48.70 | 44.44 | 61.91 | 55.73 | 45.62 | 40.87 |
| iCaRL *w S* (Ours) | 68.32 | 59.34 | 54.77 | 43.67 | 63.78 | 55.89 | 50.74 | 37.45 |
| iCaRL *w H* (Ours) | **69.47** | **60.00** | **56.21** | **45.23** | **64.13** | **57.58** | **51.02** | **41.60** |
| BEEF | 75.21 | 67.73 | 68.17 | 53.49 | 68.57 | 69.27 | 61.56 | 60.90 |
| BEEF *w S* (Ours) | 76.42 | 69.23 | 67.97 | 57.25 | 74.64 | 70.50 | 68.61 | 59.36 |
| BEEF *w H* (Ours) | **76.66** | **71.04** | **68.48** | **59.62** | **76.31** | **73.25** | **70.52** | **61.80** |
| FOSTER | 78.15 | 75.00 | 70.76 | 65.12 | 73.34 | 69.51 | 65.80 | 60.30 |
| FOSTER *w S* (Ours) | 77.73 | 73.51 | 70.32 | 64.07 | 75.25 | 71.97 | 68.45 | 62.25 |
| FOSTER *w H* (Ours) | **78.79** | **76.01** | **70.93** | **66.63** | **75.70** | **74.37** | **68.53** | **64.95** |

## 5.3 Impact of the Ratio in Hybrid Memory

We also study the impact of varying the ratio of synthetic exemplars in the hybrid memory under a limited exemplar buffer size. Using grid search, we empirically determine the optimal ratio of synthetic exemplars in the hybrid memory. In this experiment, we apply our hybrid memory to the replay-based CIL method iCaRL on CIFAR-100, using the zero-base 10-phase setting with 20 exemplars per class, and evaluate its performance at different synthetic-to-real exemplar ratios. As shown in Fig. 4, the performance peaks when the ratio is 0.5. This result suggests that synthetic exemplars and real exemplars are equally important, with both contributing to improved performance.

## 5.4 Effectiveness of Conditional Real Data Selection

To verify the effectiveness of our proposed conditional real data selection, we compared it with random sampling. Specifically, we apply the conditional real data selection in Eq.4 to choose optimal

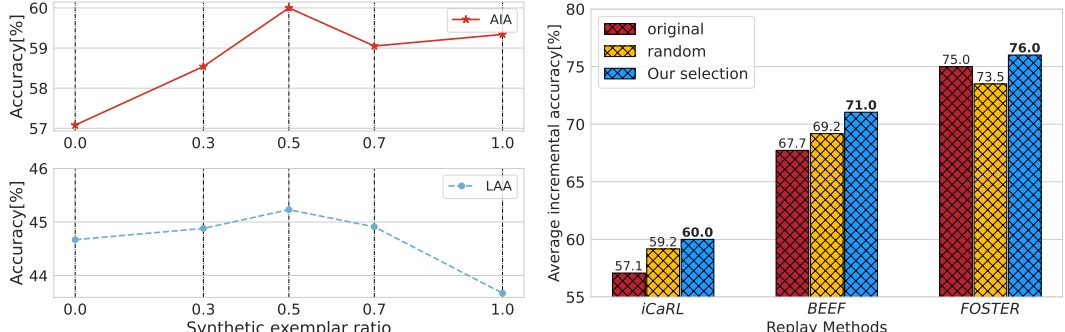

Figure 4: LAA and AIA of iCaRL with proposed hybrid memory at different synthetic exemplar ratios. "LAA" refers to the last average accuracy, "AIA" refers to the average incremental accuracy.

Figure 5: Performance comparison of the hybrid memory using different selection methods with iCaRL, BEEF, and FOSTER on CIFAR-100, all using the same exemplar buffer size.

real exemplars based on the synthetic exemplars by CDD, and then combine selected real exemplars with synthetic ones to construct a hybrid memory. For a fair comparison, we also construct a hybrid memory by randomly selecting real exemplars and combining them with synthetic exemplars. These hybrid memories are individually integrated into different replay-based CIL methods, iCaRL, FOSTER, and BEEF. We evaluate the performance using AIA on CIFAR-100 under a zero-base setting with 10 phases, with each class containing 20 exemplars. The hybrid memory maintains a balance of 10 synthetic and 10 real exemplars, while the original method utilizes 20 real exemplars. As shown in Fig.5, the hybrid memory with our conditional real data selection not only outperforms the original method but also demonstrates a significant advantage over hybrid memories with random selection. Notably, the performance comparison of FOSTER with different memory types reveals that random sampling can affect the effectiveness of the generated synthetic exemplars.The superior performance of our proposed hybrid memory is attributed to the conditional real data selection's ability to optimally choose real exemplars that complement the synthetic exemplars.

# 6 CONCLUSION AND LIMITATIONS

In this work, we introduced a hybrid memory that stores a limited number of real and synthetic exemplars for improving the CIL performance. Specifically, we devised a new Continual Data Distillation (CDD) technique that can adapt most existing DD methods into CIL to generate optimal synthetic exemplars for the general replay-based CIL models. To further complement synthetic data, we proposed a conditional real-data selection to choose optimal real exemplars based on synthetic exemplars. As a result, the combination of synthetic and real exemplars significantly enhanced the CIL performance. Extensive experimental results on two benchmarks demonstrated that our hybrid memory can improve the performance of the original methods and other baselines. A limitation of our method is that the ratio between synthetic and real exemplars in the hybrid memory is currently set as an empirical hyper-parameter. In future work, we plan to develop an adaptive algorithm to automatically determine the optimal ratio for hybrid memory.

## ETHICS STATEMENT

In this section, we address potential concerns of our study. This study focuses on Incremental learning, which aims to acquire new knowledge from current tasks while retaining knowledge learned from previous tasks. Our research complies with all relevant guidelines and regulations.

**Data Set Usage and Release.** Our study uses publicly available datasets that are widely accepted and used in the community, including CIFAR-100 and TinyImageNet. These datasets were obtained in compliance with all relevant guidelines, and they do not contain personally identifiable information.

**Privacy and Security Issues.** Our study utilizes publicly available datasets, such as CIFAR-100 and TinyImageNet, that are widely recognized and used within the research community. The data used do not contain any personally identifiable information, and the research adheres to all relevant ethical standards and privacy regulations.

**Research Integrity.** Our study follows well-established ethical guidelines for research integrity. No experiments involved human participants, and Institutional Review Board (IRB) approval was not required. All experimental methods and results are described transparently to ensure reproducibility.

**Conflict of Interest.** We declare that there is no conflict of interest regarding this work. Our research was conducted independently, without any external influence that could affect the integrity of the findings.

## REPRODUCIBILITY STATEMENT

We made efforts to ensure that our work is reproducible. The following elements in our paper and supplementary materials aim to facilitate the reproducibility of the results.

**Code Availability.** The source code for our experiments, including model training, evaluation scripts, and hyper-parameter settings, is provided. An anonymous downloadable link is available in the Abstract, enabling readers to replicate all experiments detailed in the paper.

**Detailed Experimental Settings.** We provide a complete description of the experimental setup, including model details, training strategies, evaluation metrics, and specific hyper-parameters for each experiment, in Sec. 5.

**Theoretical Assumptions and Proofs.** We provide detailed theoretical assumptions in Sec. 4.3 and corresponding proofs in Appendix C. Clear explanations of the underlying assumptions and all steps required to validate our claims are included.

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

## A  NOTATION LIST.

Table 4 describes the important notations used in our work.

Table 4: Summary of Notations.

| Notation | Definition |
|---|---|
| $t$ | The current task ID |
| $\mathcal{R}_{1:t} \triangleq \bigcup_{i=1}^{t-1} \mathcal{R}_i \cup \mathcal{R}_t$ | The real data for tasks from 1 to $t-1$ |
| $\mathcal{R}_t \triangleq \{(x_i, y_i)\}_{i=1}^{n_t}$ | The real data for task-$t$ |
| $x_i$ | The real data sample |
| $y_i$ | The label of the real data sample $x_i$ |
| $n_t$ | The number of real instances in task-$t$ |
| $C_t$ | The whole label space for task-$t$ |
| $\mathcal{H}_{1:t-1} \triangleq \bigcup_{i=1}^{t-1} \mathcal{H}_i$ | The hybrid memory for tasks from 1 to $t-1$ |
| $\mathcal{H}_t \triangleq \{\mathcal{S}_t, \mathcal{R}_t[A_t]\}$ | The hybrid memory for the task-$t$ |
| $\mathcal{S}_t$ | The synthetic exemplars for the task-$t$ |
| $\mathcal{S}_{t,j}$ | The updated synthetic exemplars for the task-$t$ after epoch-$j$ of task-$t$ in sliding window version of CDD |
| $\tau$ | The window size (the number of cached checkpoints) for sliding window version of CDD |
| $\eta$ | The learning rate for updating the synthetic exemplars $\mathcal{S}_{t,j}$ by sliding window version of CDD |
| $A_t \subseteq \{1, 2, \cdots, |\mathcal{R}_t|\}$ | The indexes of subset chosen from the real dataset $\mathcal{R}_t$ |
| $f(\cdot; \theta)$ | The whole model, with parameters $\theta$ |
| $\theta_{\mathcal{H}_t}$ | The optimal model trained on task-$t$'s hybrid memory |
| $\theta_{\mathcal{R}_t}$ | The optimal model trained on task-$t$'s real data |
| $\theta_{\mathcal{R}_{1:t}}$ | The optimal model trained on the real data of all tasks from 1 to $t$ |
| $\theta_{\mathcal{H}_{1:t}}$ | The optimal model trained on the hybrid memory of all prior tasks from 1 to $t$ |
| $\epsilon_t$ | The scalar value bounding the performance of $\theta_{\mathcal{H}_{1:t}}$ on $\mathcal{R}_{1:t}$ by the performance of $\theta_{\mathcal{R}_{1:t}}$ on $\mathcal{R}_{1:t}$. |
| $\rho$ | The scaling factor that bounds the sum of the performance of $\theta_{\mathcal{H}_{1:t}}$ on $\mathcal{R}_{1:t}$ and the performance of $\theta_{\mathcal{H}_{t+1}}$ on $\mathcal{R}_{t+1}$. |

## B  ALGORITHM OF CONDITIONAL REAL DATA SELECTION

As described in Sec. 4.3, we employ a greedy algorithm for our proposed conditional real data selection. The details of this algorithm are provided below,

---
**Algorithm 2:** Conditional Real Data Selection (Greedy Algorithm)

---
**input**   : Real data for classes $C_t$: $\mathcal{R}_t$; synthetic exemplars: $\mathcal{S}_t$; number of selected exemplars per class: $k$
**initialize** : Selected subset $A_t \leftarrow \emptyset$, minimum distance $d_{min} \leftarrow +\infty$
1 **while** $|A_t| < |C_t| \times k$ **do**
2     **for** $i \in V_t \backslash A_t$ **do**
3        $A_t' \leftarrow A_t \cup \{i\}$;
4        compute the objective value in Eq. 4 as $d$, where $A_t$ is set to $A_t'$;
5        **if** $d < d_{min}$ *and* $|\{j \in A_t : y_j = y_i\}| < k$ **then**
6           Update minimum distance: $d_{min} \leftarrow d$;
7           $i^* \leftarrow i$;
8     Update subset $A_t \leftarrow A_t \cup \{i^*\}$;

**output**   : Selected exemplars $\mathcal{R}_t[A_t]$

---

## C  THE PROOF FOR THEOREM 1.

Here we provide the detailed proof for Theorem 1.

**Theorem 1** (Performance Approximation). *Based on the above Assumptions 1 and 2, when $\epsilon_{t+1} \geq \frac{\rho}{1-\epsilon_t}$, we can derive that the model trained on the hybrid memory of all previous tasks achieves performance comparable to that of the model trained on the real dataset of all previous tasks,*

$$\log P(\mathcal{R}_{1:t}|\theta_{\mathcal{H}_{1:t}}) \geq (1 - \epsilon_t) \log P(\mathcal{R}_{1:t}|\theta_{\mathcal{R}_{1:t}}). \tag{6}$$

*Proof.* According to the Definition 4, the $\theta_{\mathcal{R}_{1:t}}$ is the optimal solution of $r_t(\theta)$, thus we can get $r_t(\theta_{\mathcal{R}_{1:t}}) \geq r_t(\theta_{\mathcal{R}_{1:t+1}})$. And according to the Definition 2, the $\theta_{\mathcal{R}_{t+1}}$ is the optimal solution of $\log P(\mathcal{R}_{t+1}|\theta)$, thus we can get $\log P(\mathcal{R}_{t+1}|\theta_{\mathcal{R}_{t+1}}) \geq \log P(\mathcal{R}_{t+1}|\theta_{\mathcal{R}_{1:t+1}})$.

According to the Mathematical Induction:

**Base Case ($t = 1$):**

$$\log P(\mathcal{R}_{1:1}|\theta_{\mathcal{H}_{1:1}}) = \log P(\mathcal{R}_1|\theta_{\mathcal{H}_1}), \ \ \log P(\mathcal{R}_{1:1}|\theta_{\mathcal{R}_{1:1}}) = \log P(\mathcal{R}_1|\theta_{\mathcal{R}_1})$$

$$\exists \epsilon \in [0, 1), \ \ \log P(\mathcal{R}_1|\theta_{\mathcal{H}_1}) \geq (1 - \epsilon) \log P(\mathcal{R}_1|\theta_{\mathcal{R}_1})$$

According to the Assumption 1, the base case holds true.

**Inductive Hypothesis:** Assume that the formula holds for $\epsilon_t$ at task-$t$:

$$\log P(\mathcal{R}_{1:t}|\theta_{\mathcal{H}_{1:t}}) \geq (1 - \epsilon_t) \log P(\mathcal{R}_{1:t}|\theta_{\mathcal{R}_{1:t}})$$

**Inductive Step:** We will prove that the formula holds for $\epsilon_{t+1}$ at task-$t + 1$:

$$\begin{aligned}
\log P(\mathcal{R}_{1:t+1}|\theta_{\mathcal{H}_{1:t+1}}) &= \log P(\mathcal{R}_{1:t}|\theta_{\mathcal{H}_{1:t+1}}) + \log P(\mathcal{R}_{t+1}|\theta_{\mathcal{H}_{1:t+1}}) \\
&= r_t(\theta_{\mathcal{H}_{1:t+1}}) + \log P(\mathcal{R}_{t+1}|\theta_{\mathcal{H}_{1:t+1}}) \\
&= r_{t+1}(\theta_{\mathcal{H}_{1:t+1}})
\end{aligned}$$

$$\begin{aligned}
\log P(\mathcal{R}_{1:t+1}|\theta_{\mathcal{R}_{1:t+1}}) &= \log P(\mathcal{R}_{1:t}|\theta_{\mathcal{R}_{1:t+1}}) + \log P(\mathcal{R}_{t+1}|\theta_{\mathcal{R}_{1:t+1}}) \\
&= r_t(\theta_{\mathcal{R}_{1:t+1}}) + \log P(\mathcal{R}_{t+1}|\theta_{\mathcal{R}_{1:t+1}}) \\
&= r_{t+1}(\theta_{\mathcal{R}_{1:t+1}})
\end{aligned}$$

$$\begin{aligned}
r_{t+1}(\theta_{\mathcal{R}_{1:t+1}}) - r_{t+1}(\theta_{\mathcal{H}_{1:t+1}}) &\leq r_t(\theta_{\mathcal{R}_{1:t+1}}) + \log P(\mathcal{R}_{t+1}|\theta_{\mathcal{R}_{1:t+1}}) \\
&\leq r_t(\theta_{\mathcal{R}_{1:t}}) + \log P(\mathcal{R}_{t+1}|\theta_{\mathcal{R}_{t+1}}) \\
&\leq \frac{1}{1 - \epsilon_t} \left( r_t(\theta_{\mathcal{H}_{1:t}}) + \log P(\mathcal{R}_{t+1}|\theta_{\mathcal{H}_{t+1}}) \right) \\
&\leq \frac{\rho}{1 - \epsilon_t} r_{t+1}(\theta_{\mathcal{R}_{1:t+1}}) \\
&\leq \epsilon_{t+1} r_{t+1}(\theta_{\mathcal{R}_{1:t+1}})
\end{aligned}$$

Thus, by the principle of mathematical induction, the formula holds for all $\epsilon_t$ of task-$t$. $\qquad\square$

**Discussion.** We discuss the performance boundary of the model trained with the proposed hybrid memory. According to Theorem 1, the performance of the optimal model trained on the hybrid memory for all tasks which is denoted as $r_t(\theta_{1:t}^H)$, is bounded by the performance of the optimal model trained on the real dataset for all tasks, scaled by $1 - \epsilon_t$, when $\epsilon_{t+1} \geq \frac{\rho}{1-\epsilon_t}$. It is evident that $\epsilon_t$ can converge to $\epsilon_t = \frac{1 \pm \sqrt{1-4\rho}}{2}$ if $\rho \leq \frac{1}{4}$, ensuring that $\epsilon_t$ remains within the bounds $\epsilon_t \in [0, 1)$. Therefore, if the initial $\epsilon_t \leq \frac{1 \pm \sqrt{1-4\rho}}{2}$, the performance will be well bounded. We also illustrate how $\epsilon_t$ changes iteratively over 100 tasks with different values of $\rho$ and initial $\epsilon_t$ in Fig. 6.

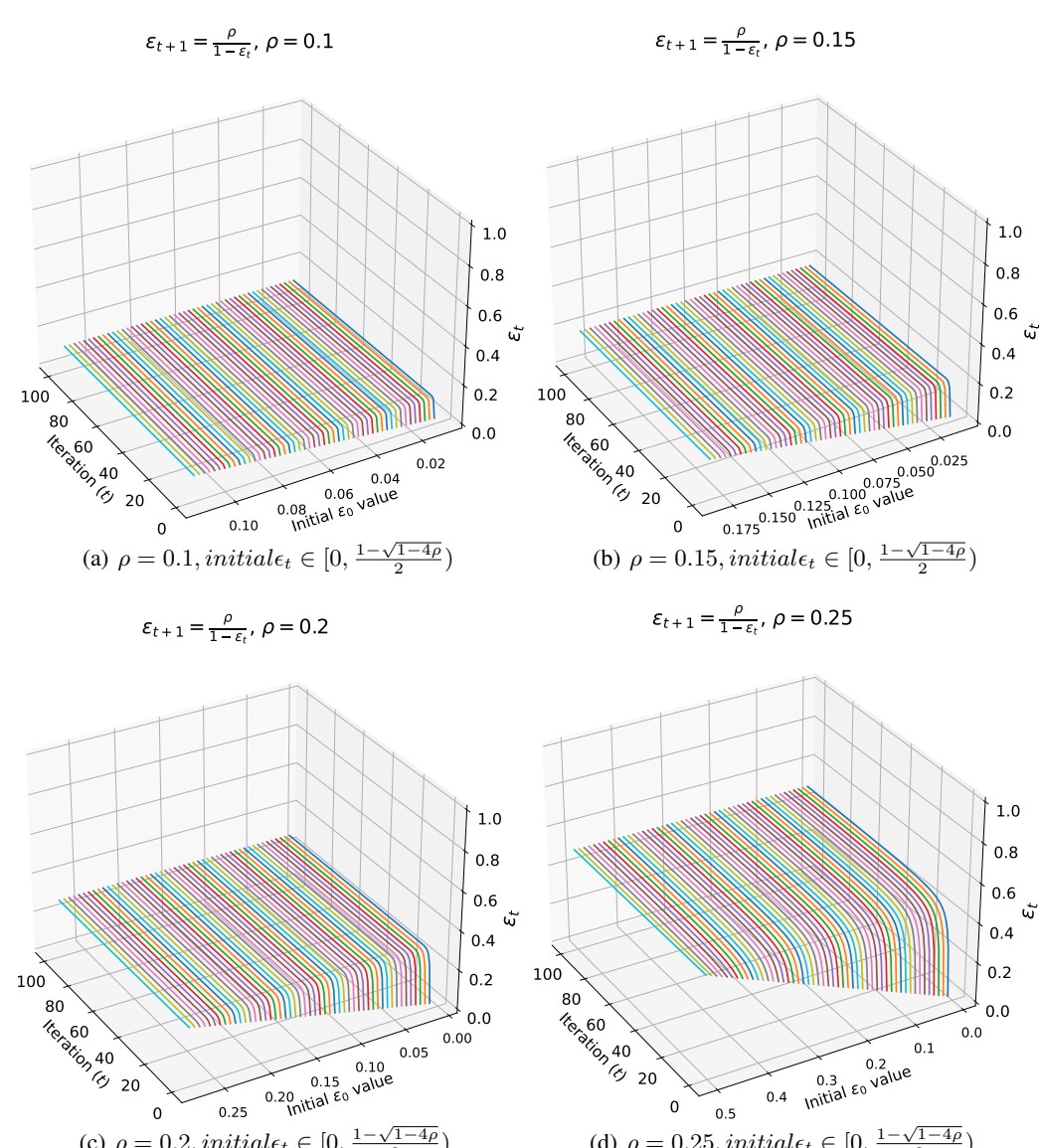

Figure 6: Visualization of $\epsilon_t$ in 100 iteration with different $\rho$ and different initial $\epsilon_t$.

## D CDD OPTIMIZATION OBJECTIVES.

We present a detailed description of several optimization objectives for our CDD approach.

**CDD with DM.** First, we describe the adaptation of DM Zhao & Bilen (2023) as the optimization objective for CDD, as outlined below,

$$\mathop{\mathbb{E}}_{\theta \sim \theta_{1:N}} [\ell_{DM}(\mathcal{S}_t, \mathcal{R}_t; \theta)] = \mathop{\mathbb{E}}_{\theta \sim \theta_{1:N}} \left[ \| \mathbb{E}[\psi(\mathcal{S}_t; \theta)] - \mathbb{E}[\psi(\mathcal{R}_t; \theta)] \|^2 \right], \tag{8}$$

where $N$ in Eq. 8 represents the total number of cached checkpoints $\theta$ in the history of learning task-$t$, $\psi(\cdot; \theta)$ is the function to extract the embedding of the input samples. The objective of Eq. 8 aims to measure the distance between the mean vector of the synthetic exemplars and the mean vector of the real dataset.

**CDD with FTD.** We also adapt the FTD Du et al. (2023) as the optimization objective for CDD as follows,

$$
\mathbb{E}_{\theta \sim \theta_{1:N}} [\ell_{FTD}(\mathcal{S}_t, \{\mathcal{R}_t, \mathcal{H}_{1:t-1}\}; \theta)] = \mathbb{E}_{\theta \sim \theta_{1:N}} \frac{\|(\theta_S^{(n)} - \theta^{(m)})\|_2^2}{\|(\theta - \theta^{(m)})\|_2^2},
$$
$$
\theta_S^{(0)} = \theta, \ \theta_S^{(n)} = \theta_S^{(n-1)} + \eta \nabla L(\mathcal{S}_t; \theta_S^{(n-1)}),
$$
$$
\theta^{(0)} = \theta, \ \theta^{(m)} = \theta^{(m-1)} + \eta \nabla L(\{\mathcal{R}_t, \mathcal{H}_{1:t-1}\}; \theta^{(m-1)}), \tag{9}
$$

where $N$ in Eq. 9 represents the total number of cached checkpoints $\theta$ in the history of learning task-$t$. The parameter vectors $\theta_S^{(n)}$ and $\theta^{(m)}$, which originate from the initial model parameters $\theta$, are iteratively trained on the synthetic dataset $\mathcal{S}_t$ and the data $\mathcal{R}_t \cup \mathcal{H}_{1:t-1}$ of the current task, respectively, for $n$ and $m$ steps with learning rate $\eta$. The function $L(\cdot; \cdot)$ is the loss function used for training the network across different CIL methods. The objective in Eq. 9 focuses on quantifying the difference between two sets of parameter vectors: $\theta_S^{(n)}$ and $\theta^{(m)}$. Here $\theta_S^{(n)}$ is derived from the initial parameters $\theta$, which are trained with synthetic exemplars $\mathcal{S}_t$ for $n$ steps, while $\theta^{(m)}$ originates from the same initial parameters but is trained with real data $\mathcal{R}_t \cup \mathcal{H}_{1:t-1}$ for $m$ steps.

**CDD with DSA.** Additionally, we also adapt the DSA Zhao & Bilen (2021) as the optimization objective for CDD as follows,

$$
\mathbb{E}_{\theta \sim \theta_{1:N}} [\ell_{DSA}(\mathcal{S}_t, \mathcal{R}_t; \theta)] = \mathbb{E}_{\theta \sim \theta_{1:N}} \frac{< \nabla L(\mathcal{S}_t; \theta), \nabla L(\mathcal{R}_t; \theta) >}{\|\nabla L(\mathcal{S}_t; \theta)\|_2, \|\nabla L(\mathcal{R}_t; \theta)\|_2}, \tag{10}
$$

where $N$ in Eq. 10 represents the total number of cached checkpoints $\theta$ in the history of learning task-$t$. The function $L(\cdot; \cdot)$ is the loss function used for training the network across different CIL methods. The objective in Eq. 10 focuses on quantifying the cosine similarity between the gradient of the synthetic exemplars $\mathcal{S}_t$ and the gradient of the real dataset $\mathcal{R}_t$, both computed on the sampled parameters $\theta$ with the loss function $L(\cdot; \cdot)$.

**CDD with DataDAM.** Furthermore, we adapt the DataDAM Sajedi et al. (2023) as the optimization objective for CDD as follows,

$$
\mathbb{E}_{\theta \sim \theta_{1:N}} [\ell_{DataDAM}(\mathcal{S}_t, \mathcal{R}_t; \theta)] = \mathbb{E}_{\theta \sim \theta_{1:N}} [L_{SAM}(\mathcal{S}_t, \mathcal{R}_t; \theta) + \lambda L_{MMD}(\mathcal{S}_t, \mathcal{R}_t; \theta)], \tag{11}
$$

where $N$ in Eq. 10 represents the total number of cached checkpoints $\theta$ in the history of learning task-$t$. The objective in Eq. 11 focuses on attention matching and quantifying the distributions between synthetic exemplars and real datasets by objectives $L_{SAM}(\cdot, \cdot; \cdot)$ and $L_{MMD}(\cdot, \cdot; \cdot)$.

Furthermore, we validate the wild applicability of our CDD in adapting the data distillation methods into CIL. In this experiment, we adapt different DD algorithms, DM, FTD, DSA, and DataDAM above as CDD optimization objectives. These objectives with CDD are integrated into iCaRL, and their performance is evaluated on the CIFAR-100 dataset with zero-base 10 phases setting. In the experiment, each CDD is used to generate **10** synthetic exemplars per class, denoted as $S(DM)$, $S(FTD)$, $S(DSA)$, and $S(DataDAM)$. These are compared against the original iCaRL method, which selects 10 and 20 real exemplars per class using *herding* Welling (2009), as described in Rebuffi et al. (2017). As illustrated in Fig. 7, iCaRL with 10 synthetic exemplars per class outperforms the original method which uses 10 real exemplars per class (black curve). In addition, using just

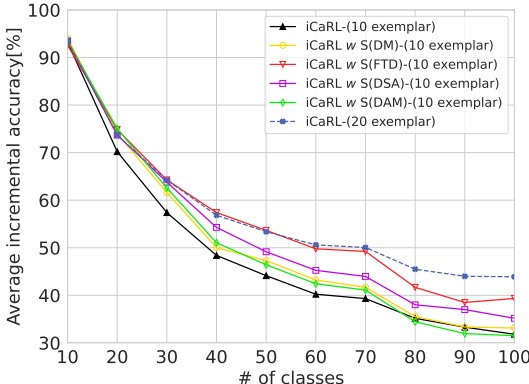

Figure 7: Results of different CDD optimization objectives on CIFAR-100 with zero-base 10 phases setting. S($\cdot$) means synthetic memory generated by a specific objective.

10 synthetic exemplars per class as $S(FTD)$ achieves comparable performance to iCaRL configured with 20 real exemplars per class (blue dash curve).

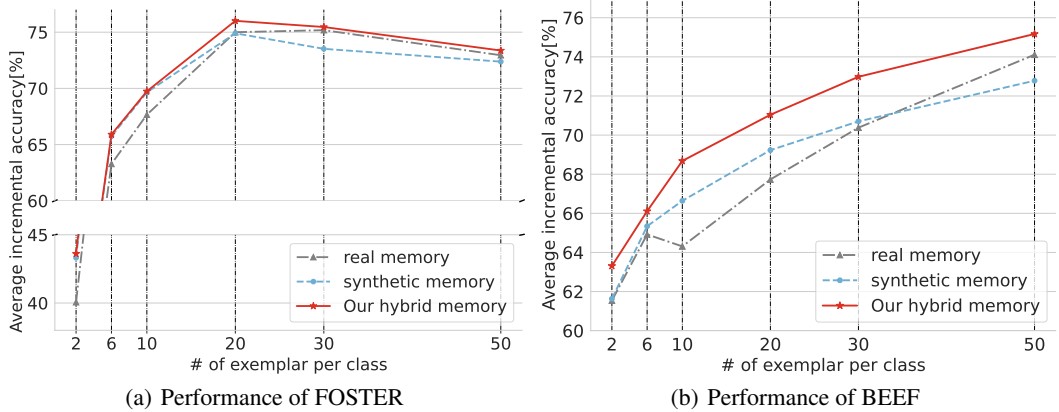

(a) Performance of FOSTER  (b) Performance of BEEF

Figure 8: Performance evaluations of FOSTER and BEEF across different exemplar buffer sizes for real memory, synthetic memory, and our hybrid memory. "real memory" refers to buffers containing only real exemplars selected by iCaRL. "synthetic memory" contains only synthetic exemplars generated using CDD.

## E  EFFECTIVENESS OF HYBRID MEMORY ACROSS DIFFERENT EXEMPLAR BUFFER SIZES

To further validate the assumption from Sec. 3 that synthetic exemplars alone are not always sufficient, we explore the effectiveness of the hybrid memory across different exemplar buffer sizes with several different replay-based CIL methods. In addition to the iCaRL results shown in Fig. 2, we compare the performance of hybrid memory with both synthetic and real memory on two other replay-based CIL methods, BEEF and FOSTER, across various buffer sizes. The experiments were conducted on CIFAR-100 under the zero-base 10-phase setting, the same setting used for iCaRL in Fig. 2.

As shown in Fig. 8, the performance of synthetic memory generated by CDD significantly improves at smaller buffer sizes but quickly diminishes as the number of synthetic exemplars increases in both FOSTER and BEEF. In contrast, our proposed hybrid memory leverages the strengths of both synthetic and real exemplars, consistently outperforming synthetic and real memory across different exemplar buffer sizes.

## F  ADDITIONAL RESULTS

To demonstrate the superior performance of our approach, we also compare the average accuracy curves for each task against the baselines. As described in Sec. 5.1, we here present the average accuracy (AA) for each task corresponding to Table 1 and Table 2. As shown in Fig. 9, a comparison between the solid curves (representing methods with our proposed hybrid memory) and the dashed curves of the same color (representing the original methods) shows that our hybrid memory consistently enhances the performance of replay-based CIL methods, even with a limited exemplar buffer size, across different dataset scales and experimental settings.

## G  VISUALIZATION

In this section, we provide visualizations of the synthetic and selected exemplars from CIFAR-100 and TinyImageNet. As shown in Fig. 10, we randomly select three different classes and compare the synthetic exemplars generated by our CDD with the corresponding selected exemplars conditioned on these synthetic exemplars. By comparing the upper and bottom rows, we observe that the selected exemplars are chosen based on the given synthetic exemplars and provide complementary information that the synthetic exemplars alone do not capture.

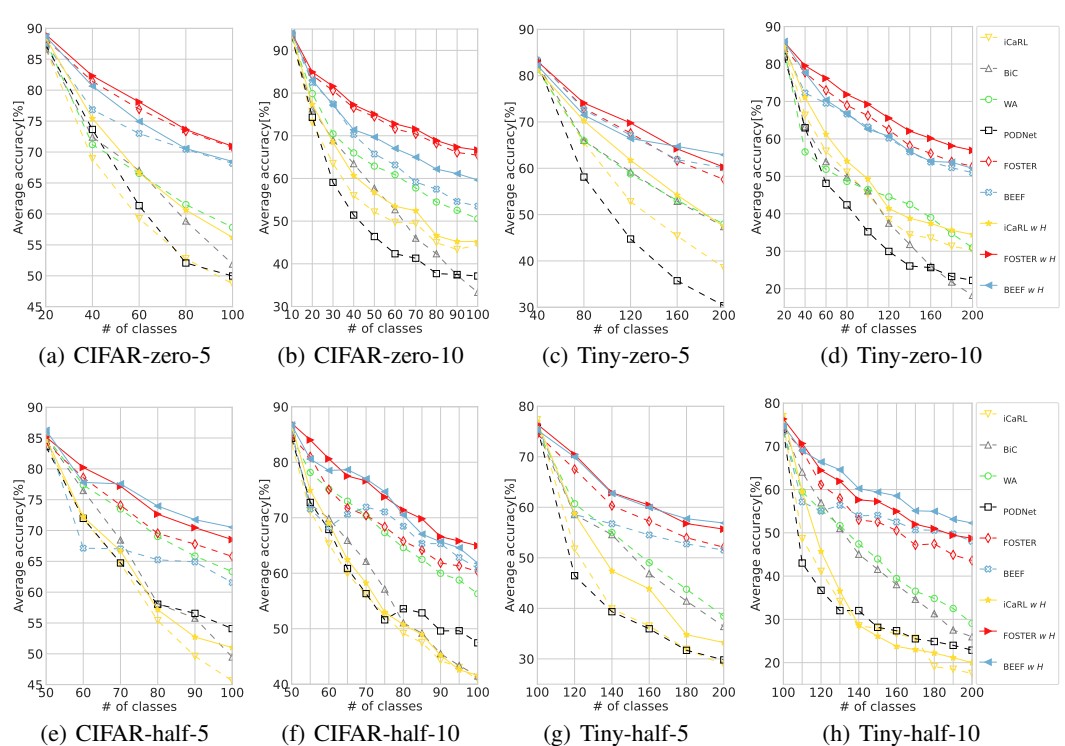

Figure 9: Accuracy comparison of different methods on CIFAR-100 and TinyImageNet under different settings. It can be observed that our hybrid memory can improve the performance of repla-based methods in terms of accuracy. "zero-5,10" means "zero-base-5,10 phases" setting and "half-5,10" means "half-base-5,10 phases" setting.

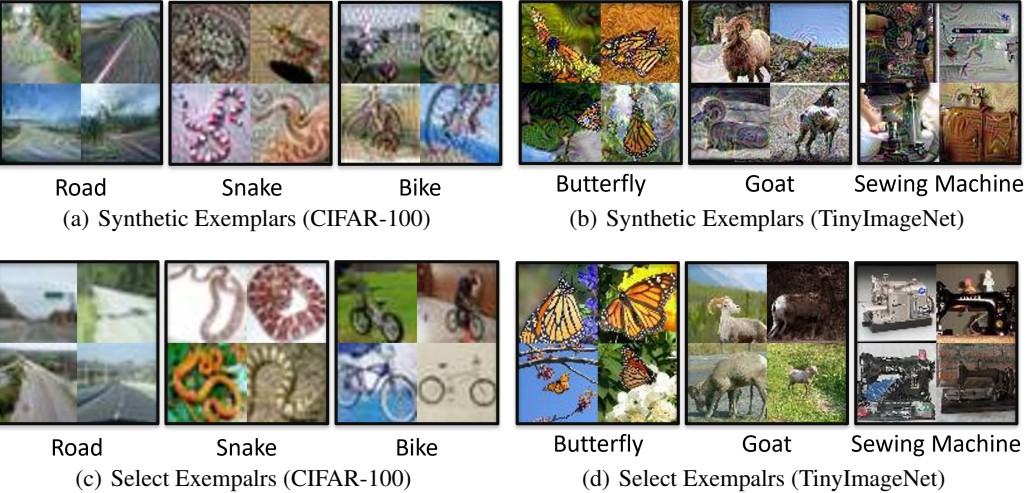

Figure 10: Visualization of select exemplars and synthetic exemplars, the latter generated by CDD with DM, from CIFAR-100 and TinyImageNet.

