# OpenReview forum: "Hybrid Memory Replay: Blending Real and Distilled Data for Class Incremental Learning"
_ICLR.cc/2025/Conference — ICLR 2025 Conference Withdrawn Submission_

### Official Review · Reviewer_bVTA · 2024-11-02

**Soundness:** 3
**Presentation:** 3
**Contribution:** 2
**Rating:** 3
**Confidence:** 4

**Summary:**

This paper proposes a new hybrid memory system for replay-based CIL by combining both synthetic and real exemplars. The authors propose a new data distillation method, namely Continual Data Distillation (CDD), which adapts data distillation for CIL by extracting synthetic exemplars without requiring multiple training checkpoints. Their experiments show that while synthetic-only memory initially performs better, its effectiveness diminishes as the memory buffer grows. The authors then propose a conditional real data selection method, by selecting optimal real samples to complement synthetic ones. Their experiment results show that their hybrid approach outperforms the method using only real exemplars.

**Strengths:**

* The authors’ proposed method of combining real and synthetic (distilled) data for CIL is interesting and might be potentially useful for certain scenarios.
* Their proposed method seems to be integrated into (some) existing replay-based CIL methods.

**Weaknesses:**

* The novelty seems to be limited: the proposed methods looks rather straightforward. Instead of jointly optimizing the optimal sets of real and synthetic data, the authors first simply construct synthetic samples (without worrying about the real samples to be selected later) and then, fixing the synthetic samples, they now simply select real samples based on a greedy algorithm. Although the authors claim that using a sliding window of checkpoints is an innovative modification, it also seems rather an obvious (trivial) modification.
* There is no guarantee or evidence that the two assumptions, Assumption 1 and Assumption 2, are true. Therefore, it is not possible if Theorem 1 is valid or not. In this case, I really do not know if we can call it a theorem.
* The hybrid memory is composed of equal numbers of real and synthetic samples. But, I do not know why they must be the same. The ratio between those two groups of samples seems to be a critical parameter to be optimized. Fig 4 might support the equal numbers. But, this is a very simple simulation for a very specific setting for iCaRL. How about other more recent and advanced IL methods across various datasets?
* Recently, more and more researchers in this area (including the papers for both FOSTER and BEEF that the authors considered) test ImageNet-1k as well in their simulation, which is not the case in this paper.

**Questions:**

* “However, as the buffer size increases, synthetic exemplars become less effective than real exemplars, likely due to the inherent important information loss in the distillation process” I do not know if this is true only for the very particular simulation setting(s) considered in this paper.  Or, if it is universally true. Are there any other publications that can support this claim, especially in the area of data distillation? I am also not sure about “likely due to the inherent important information loss in the distillation process…” If a data distillation method loses “important” information, the method was not designed well. The objective of all data distillation methods should be keeping “important” information. Also, as the number of synthetic exemplars grows, the loss of “important” information must be less and less by any reasonably designed data distillation methods. Therefore, “likely due to the inherent important information loss in the distillation process” doesn’t explain why “as the buffer size increases, synthetic exemplars become less effective than real exemplars”.
* To solve (1), the authors use (2) (or (3)) and (4), which looks straightforward. Is this the most efficient approach to solve (1)? Are there any other approaches that can be potentially better?
* To solve (4), greedy algorithms are used. Any other more effective methods?
* Isn't it possible at all to prove Assumption 1 and Assumption 2? If not, how can we trust the results of Theorem 1? At the very least, do we have any numerical results to support those assumptions?
* The numerical results of FOSTER in this paper do not seem to match those in the original paper of FOSTER. Also, for BEEF (including BEEF-Compress), the numerical results in the original paper seem to be better than those in this paper.
• Any experiment results on ImageNet-1k?

---

### Official Review · Reviewer_neoy · 2024-11-03

**Soundness:** 3
**Presentation:** 3
**Contribution:** 3
**Rating:** 5
**Confidence:** 3

**Summary:**

The authors propose a modified data distillation method to develop a hybrid memory approach that combines real and synthetic examples, to reduce catastrophic forgetting in Class Incremental Learning (CIL) without requiring extensive data storage. This hybrid memory relies on a novel technique called Continual Data Distillation (CDD), which generates synthetic exemplars from a sequence of checkpoints taken along the same model’s training trajectory, instead of needing multiple model initializations. The work mainly builds on a memory update strategy, where real data are conditionally selected to complement synthetic exemplars, maximizing a data distillation objective through a greedy selection process.

**Strengths:**

The motivation for this work is strong, clear, and easy to follow. By addressing the challenge of catastrophic forgetting in Class Incremental Learning (CIL), the authors offer a practical solution that optimizes memory use while maintaining performance.

Several studies, including theoretical discussions, ablation studies on the update strategy, investigations into the synthetic/real exemplar ratio, and other relevant analyses, reinforce the validity of this work.

**Weaknesses:**

- Instead of focusing on reporting results for both the **zero-base** setting and the **half-base** setting- which I consider less important—the authors could have presented results on other benchmarks, particularly since using only two datasets is rather limited. Expanding the range of benchmarks, especially across slightly different domains, *e.g.*, Caltech256 (*Griffin, G. Holub, AD. Perona, P*) or MIT67 (*Quattoni & Torralba*), would enhance the relevance of their findings.

- While the authors adhere to the standards set by other works, it would be beneficial to provide explicitly how **AIA** and **LAA** are computed. This would facilitate a better understanding of the results. Could you please include a brief description of the AIA and LAA computations either in the main text or as a footnote in the results tables?

- I recommend including the standard upper and lower bounds for Class Incremental Learning (*i.e.*, training on all tasks jointly and finetuning with no countermeasure to forgetting). This would provide a more comprehensive view of the method's capabilities.

**Questions:**

I would like to confirm whether the results presented in the base version of the competitor methods (*i.e.*, the upper part of Table 1) are derived using their original replay mechanisms, such as iCaRL with Herding, and not altered in any way. Additionally, could you clarify how these original methods have been combined with your proposed approach, particularly in relation to the introduction of Hybrid Memory (*i.e.*, the lower part *w. H*)?

Have you considered whether the success of your data distillation might be sufficient when paired with a strategy that maximizes diversity? For example, could already existing Memory Update techniques (like Herding or the approach based on Algo. 2 of [1]) make your Hybrid Memory effective without requiring conditional updates for the real data? This analysis could provide valuable insights and enrich the discussion presented in Figure 5.

     [1] Buzzega, Pietro, et al. "Rethinking experience replay: a bag of tricks for continual learning." 2020 25th International Conference on Pattern Recognition (ICPR). IEEE, 2021.

I noticed that you mentioned the results are averaged over three runs. It would be helpful to include the standard deviations at least in the main results Tables.

**Details Of Ethics Concerns:**

No Ethics Concerns.

---

### Official Review · Reviewer_qTie · 2024-11-08

**Soundness:** 3
**Presentation:** 2
**Contribution:** 2
**Rating:** 3
**Confidence:** 2

**Summary:**

The authors propose a new approach to incremental learning called "Hybrid Memory", which combines the benefits of replay-based methods with leveraging both real exemplars and distilled data, which can potentially reduce the exemplar buffer size while mitigating catastrophic forgetting. Their experimental results show that Hybrid Memory outperforms existing replay-based CIL methods on several benchmarks, including CIFAR-100 and Tiny ImageNet.

**Strengths:**

- The approach of using real and distilled data within a hybrid memory framework is interesting.
- The proposed hybrid memory can be integrated seamlessly with existing replay-based CIL methods like iCaRL, BEEF, and FOSTER, making it a versatile addition to existing frameworks.
- Availability of code is appreciated.

**Weaknesses:**

- The authors said, "We do not need multiple trajectories because the replay of St in the next task will only start from θN instead of random initializations." As you may know the reason to use multiple training trajectories is to gain more robust and representative dataset for approximate the complex structure of the original dataset, improving generalization and reducing overfitting to any single path of learning. As you didn't provide any proof over generalization of your method, it is a real concern that your model might be overfitted on dataset and achieve higher accuracy over other methods.
- The proposed method "Hybrid Memory", relies heavily on existing replay-based CIL, with minimal innovation or breakthroughs presented in this work. The authors fail to provide a clear and compelling justification for their approach, instead relying on vague claims about improving performance on certain benchmarks.
- Given the focuses of paper is on hybrid exemplars, an analysis of computational and memory overhead associated with the hybrid memory structure is essential.
- Paper needs ablation study.

**Questions:**

1- How does the synthetic-to-real exemplar ratio impact performance in tasks with more significant shifts in data distribution? Would an imbalance lead to overfitting to synthetic data, particularly in settings with high task complexity?
2- Can the authors clarify the criteria for synthetic exemplar selection beyond theoretical metrics? How does the algorithm ensure diversity in synthetic data without redundancy?

---

### Note · Authors · 2024-11-15

**Comment:**

Thank you! We will address the comments from reviewers.

**Withdrawal Confirmation:**

I have read and agree with the venue's withdrawal policy on behalf of myself and my co-authors.